# CPT: Efficient Deep Neural Network Training via Cyclic Precision

**Yonggan Fu, Han Guo, Xin Yang, Yining Ding & Yingyan Lin**
Department of Electrical and Computer Engineering
Rice University
{yf22, hg31, xy33, yd31, yingyan.lin}@rice.edu

**Meng Li & Vikas Chandra**
Facebook Inc
{meng.li, vchandra}@fb.com

## Abstract

Low-precision deep neural network (DNN) training has gained tremendous attention as reducing precision is one of the most effective knobs for boosting DNNs' training time/energy efficiency. In this paper, we attempt to explore low-precision training from a new perspective as inspired by recent findings in understanding DNN training: we conjecture that DNNs' precision might have a similar effect as the learning rate during DNN training, and advocate dynamic precision along the training trajectory for further boosting the time/energy efficiency of DNN training. Specifically, we propose Cyclic Precision Training (CPT) to cyclically vary the precision between two boundary values which can be identified using a simple precision range test within the first few training epochs. Extensive simulations and ablation studies on five datasets and eleven models demonstrate that CPT's effectiveness is consistent across various models/tasks (including classification and language modeling). Furthermore, through experiments and visualization we show that CPT helps to (1) converge to a wider minima with a lower generalization error and (2) reduce training variance which we believe opens up a new design knob for simultaneously improving the optimization and efficiency of DNN training. Our codes are available at: https://github.com/RICE-EIC/CPT.

## 1 Introduction

The record-breaking performance of modern deep neural networks (DNNs) comes at a prohibitive training cost due to the required massive training data and parameters, limiting the development of the highly demanded DNN-powered intelligent solutions for numerous applications (Liu et al., 2018; Wu et al., 2018). As an illustration, training ResNet-50 involves $10^{18}$ FLOPs (floating-point operations) and can take 14 days on one state-of-the-art (SOTA) GPU (You et al., 2020b). Meanwhile, the large DNN training costs have raised increasing financial and environmental concerns. For example, it is estimated that training one DNN can cost more than $10K US dollars and emit carbon as high as a car's lifetime emissions. In parallel, recent DNN advances have fueled a tremendous need for intelligent edge devices, many of which require on-device in-situ learning to ensure the accuracy under dynamic real-world environments, where there is a mismatch between the devices' limited resources and the prohibitive training costs (Wang et al., 2019b; Li et al., 2020; You et al., 2020a).

To address the aforementioned challenges, extensive research efforts have been devoted to developing efficient DNN training techniques. Among them, low-precision training has gained significant attention as it can largely boost the training time/energy efficiency (Jacob et al., 2018; Wang et al., 2018a; Sun et al., 2019). For instance, GPUs can now perform mixed-precision DNN training with 16-bit IEEE Half-Precision floating-point formats (Micikevicius et al., 2017b). Despite their promise, existing low-precision works have not yet fully explored the opportunity of leveraging recent findings in understanding DNN training. In particular, existing works mostly fix the model precision during the whole training process, i.e., adopt a **static** quantization strategy, while recent works in DNN training optimization suggest dynamic hyper-parameters along DNNs' training trajectory. For example, (Li

et al., 2019) shows that a large initial learning rate helps the model to memorize easier-to-fit and more generalizable patterns, which aligns with the common practice to start from a large learning rate for exploration and anneal to a small one for final convergence; and (Smith, 2017; Loshchilov & Hutter, 2016) improve DNNs' classification accuracy by adopting cyclical learning rates.

In this work, we advocate dynamic precision training, and make the following contributions:

- We show that DNNs' precision seems to have a similar effect as the learning rate during DNN training, i.e., low precision with large quantization noise helps DNN training exploration while high precision with more accurate updates aids model convergence, and dynamic precision schedules help DNNs converge to a better minima. This finding opens up a design knob for simultaneously improving the optimization and efficiency of DNN training.

- We propose Cyclic Precision Training (CPT) which adopts a cyclic precision schedule along DNNs' training trajectory for pushing forward the achievable trade-offs between DNNs' accuracy and training efficiency. Furthermore, we show that the cyclic precision bounds can be automatically identified at the very early stage of training using a simple precision range test, which has a negligible computational overhead.

- Extensive experiments on **five datasets** and **eleven models** across a wide spectrum of applications (including classification and language modeling) validate the consistent effectiveness of the proposed CPT technique in boosting the training efficiency while leading to a comparable or even better accuracy. Furthermore, we provide loss surface visualization for better understanding CPT's effectiveness and discuss its connection with recent findings in understanding DNNs' training optimization.

## 2    RELATED WORKS

**Quantized DNNs.** DNN quantization (Courbariaux et al., 2015; 2016; Rastegari et al., 2016; Zhu et al., 2016; Li et al., 2016; Jacob et al., 2018; Mishra & Marr, 2017; Mishra et al., 2017; Park et al., 2017; Zhou et al., 2016) has been well explored based on the target accuracy-efficiency trade-offs. For example, (Jacob et al., 2018) proposes quantization-aware training to preserve the post quantization accuracy; (Jung et al., 2019; Bhalgat et al., 2020; Esser et al., 2019; Park & Yoo, 2020) strive to improve low-precision DNNs' accuracy using learnable quantizers. Mixed-precision DNN quantization (Wang et al., 2019a; Xu et al., 2018; Elthakeb et al., 2020; Zhou et al., 2017) assigns different bitwidths for different layers/filters. While these works all adopt a **static** quantization strategy, i.e., the assigned precision is fixed post quantization, CPT adopts a **dynamic** precision schedule during the training process.

**Low-precision DNN training.** Pioneering works (Wang et al., 2018a; Banner et al., 2018; Micikevicius et al., 2017a; Gupta et al., 2015; Sun et al., 2019) have shown that DNNs can be trained with reduced precision. For distributed learning, (Seide et al., 2014; De Sa et al., 2017; Wen et al., 2017; Bernstein et al., 2018) quantize the gradients to reduce the communication costs, where the training computations still adopt full precision; For centralized/on-device learning, the weights, activations, gradients, and errors involved in both the forward and backward computations all adopt reduced precision. Our CPT can be applied on top of these low-precision training techniques, all of which adopt a **static** precision during the **whole** training trajectory, to further boost the training efficiency.

**Dynamic-precision DNNs.** There exist some dynamic precision works which aim to derive a quantized DNN for inference after the full-precision training. Specifically, (Zhuang et al., 2018) first trains a full-precision model to reach convergence and then gradually decreases the model precision to the target one for achieving better inference accuracy; (Khoram & Li, 2018) also starts from a full-precision model and then gradually learns the precision of each layer to derive a mixed-precision counterpart; (Yang & Jin, 2020) learns a fractional precision of each layer/filter based on the linear interpolation of two consecutive bitwidths which doubles the computation and requires an extra fine-tuning step; and (Shen et al., 2020) proposes to adapt the precision of each layer during inference in an input-dependent manner to balance computational cost and accuracy.

## 3    THE PROPOSED CPT TECHNIQUE

In this section, we first introduce the hypothesis that motivates us to develop CPT using visualization examples in Sec. 3.1, and then present the CPT concept in Sec. 3.2 followed by the Precision Range Test (PRT) method in Sec. 3.3, where PRT aims to automate the precision schedule for CPT.

Table 1: The test accuracy of ResNet-38/74 trained on CIFAR-100 with different learning rate and precision combinations in the first stage. Note that the last two stages of all the experiments are trained with full precision and a learning rate of 0.01 and 0.001, respectively.

| | **ResNet-38** | | | | **ResNet-74** | | | |
|---|---|---|---|---|---|---|---|---|
| First-stage LR | 0.1 | 0.06 | 0.03 | 0.01 | 0.1 | 0.06 | 0.03 | 0.01 |
| 4-bit Acc (%) | 69.45 | 68.63 | **67.69** | 65.90 | 70.96 | 69.54 | 68.26 | **67.19** |
| 6-bit Acc (%) | 70.22 | 68.87 | 67.15 | **66.10** | 71.62 | 70.28 | **68.84** | 66.16 |
| 8-bit Acc (%) | 69.96 | 68.66 | 66.75 | 64.99 | 71.60 | **70.67** | 68.45 | 65.85 |
| FP Acc (%) | **70.45** | **69.53** | 67.47 | 64.50 | **71.66** | 70.00 | 68.69 | 65.62 |

## 3.1 CPT: MOTIVATION

**Hypothesis 1: DNN's precision has a similar effect as the learning rate.** Existing works (Grandvalet et al., 1997; Neelakantan et al., 2015) show that noise can help DNN training theoretically or empirically, motivating us to rethink the role of quantization in DNN training. We conjecture that low precision with large quantization noise helps DNN training exploration with an effect similar to a high learning rate, while high precision with more accurate updates aids model convergence, similar to a low learning rate.

**Validating Hypothesis 1.** Settings: To empirically justify our hypothesis, we train ResNet-38/74 on the CIFAR-100 dataset for 160 epochs following the basic training setting as in Sec. 4.1. In particular, we divide the training of 160 epochs into three stages: [0-th, 80-th], [80-th,120-th], and [120-th, 160-th]: for the first training stage of [0-th, 80-th], we adopt different learning rates and precisions for the weights and activations, while using full precision for the remaining two stages with a learning rate of 0.01 for the [80-th,120-th] epochs and 0.001 for the [120-th, 160-th] epochs in all the experiments in order to explore the relationship between the learning rate and precision in the first training stage.

Results: As shown in Tab. 1, we can observe that as the learning rate is sufficiently reduced for the first training stage, adopting a lower precision for this stage will lead to a higher accuracy than training with full precision. In particular, with the standard initial learning rate of 0.1, full precision training achieves a 1.00%/0.70% higher accuracy than the 4-bit one on ResNet-38/74, respectively; whereas as the initial learning rate decreases, this accuracy gap gradually narrows and then reverses, e.g., when the initial learning rate becomes 1e-2, training with [0-th, 80-th] of 4-bit achieves a 1.40%/1.57% higher accuracy than the full precision ones.

Insights: This set of experiments show that (1) when the initial learning rate is low, training with lower initial precisions consistently leads to a better accuracy than training with full precision, indicating that lowering the precision introduces a similar effect of favoring exploration as that of a high learning rate; and (2) although a low precision can alleviate the accuracy drop caused by a low learning rate, a high learning rate is in general necessary to maximize the accuracy.

**Hypothesis 2: Dynamic precision helps DNN generalization.** Recent findings in DNN training have motivated us to better utilize DNN precision to achieve a win-win in both DNN accuracy and efficiency. Specifically, it has been discussed that (1) DNNs learn to fit different patterns at different training stages, e.g., (Rahaman et al., 2019; Xu et al., 2019) reveal that DNN training first learns lower-frequency components and then high-frequency features, with the former being more robust to perturbations and noises; and (2) dynamic learning rate schedules help to improve the optimization in DNN training, e.g., (Li et al., 2019) points out that a large

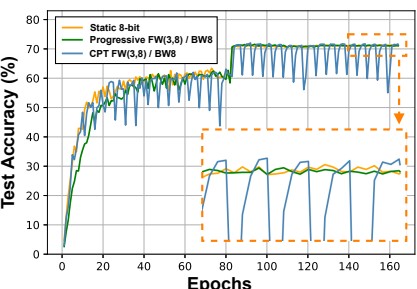

Figure 1: Test accuracy evolution of ResNet-74 on CIFAR-100 under different schedules.

initial learning rate helps the model to memorize easier-to-fit and more generalizable patterns while (Smith, 2017; Loshchilov & Hutter, 2016) show that cyclical learning rate schedules improve DNNs' classification accuracy. These works inspire us to hypothesize that dynamic precision might help DNNs to reach a better optimum in the optimization landscape, especially considering the similar effect between the learning rate and precision validated in our Hypothesis 1.

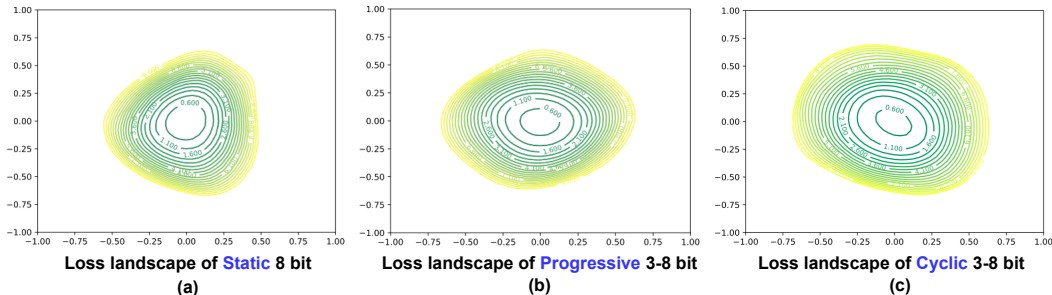

**Loss landscape of Static 8 bit** (a)      **Loss landscape of Progressive 3-8 bit** (b)      **Loss landscape of Cyclic 3-8 bit** (c)

Figure 2: Loss landscape visualization after convergence of ResNet-74 on CIFAR-100 trained with different precision schedules, where wider contours with larger intervals indicate a better local minima and a lower generalization error as analyzed in (Li et al., 2018).

**Validating Hypothesis 2.** Our Hypothesis 2 has been consistently confirmed by various empirical observations. For example, a recent work (Fu et al., 2020) proposes to progressively increase the precision during the training process, and we follow their settings to validate our hypothesis.

Settings: We train a ResNet-74 on CIFAR-100 using the same training setting as (Wang et al., 2018b) except that we quantize the weights, activations, and gradients during training; for the **progressive** precision case we uniformly increase the precision of weights and activations from 3-bit to 8-bit in the first 80 epochs and adopt static 8-bit gradients, while the static precision baseline uses 8-bit for all the weights/activations/gradients.

Results: Fig. 1 shows that training with progressive precision schedule achieves a slightly higher accuracy (+0.3%) than its static counterpart, while the former can reduce training costs. Furthermore, we visualize the loss landscape (following the method in (Li et al., 2018)) in Fig. 2(b): interestingly the progressive precision schedule helps to converge to a better local minima with wider contours, indicating a lower generalization error (Li et al., 2018) over the static 8-bit baseline in Fig. 2(a).

The progressive precision schedule in (Fu et al., 2020) relies on manual hyper-parameter tuning. As such, a natural following question would be: what kind of dynamic schedules would be effective while being simple to implement for different tasks/models? In this work, we show that a simple cyclic schedule consistently benefits the training convergence while boosting the training efficiency.

## 3.2 CPT: THE KEY CONCEPT

The key concept of CPT draws inspiration from (Li et al., 2019) which demonstrates that a large initial learning rate helps the model to learn more generalizable patterns. We thus hypothesize that a lower precision that leads to a short-term poor accuracy might actually help the DNN exploration during training thanks to its associated larger quantization noise, while it is well known that a higher precision enables the learning of higher-complexity, fine-grained patterns that is critical to better convergence. Together, this combination could improve the achieved ac-

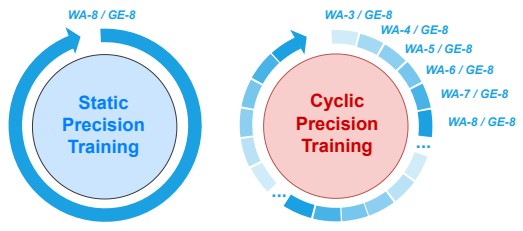

Figure 3: Static vs. Cyclic Precision Training (CPT), where CPT cyclically schedules the precision of weights and activations during training.

curacy as it might better balance coarse-grained exploration and fine-grained optimization during DNN training, which leads to the idea of CPT. Specifically, as shown in Fig. 3, CPT varies the precision cyclically between two bounds instead of fixing the precision during training, letting the models explore the optimization landscape with different granularities.

While CPT can be implemented using different cyclic scheduling methods, here we present as an example an implementation of CPT in a cosine manner:

$$B_t^n = \lceil B_{min}^n + \frac{1}{2}(B_{max}^n - B_{min}^n)(1 - cos(\frac{t \% T_n}{T_n}\pi)) \rceil \tag{1}$$

where $B_{min}^n$ and $B_{max}^n$ are the lower and upper precision bound, respectively, in the $n$-th cycle of precision schedule, $\lceil \cdot \rceil$ and $\%$ denote the rounding operation and the remainder operation, respectively,

and $B_t^n$ is the precision at the $t$-th global epoch which falls into the $n$-th cycle with a cycle length of $T_n$. Note that the cycle length $T_n$ is equal to the total number of training epochs divided by the total number of cycles denoted as $N$, where $N$ is a hyper-parameter of CPT. For example, if $N = 2$, then a DNN training with CPT will experience two cycles of cyclic precision schedule during training. As shown in Sec. 4.3, we find that the benefits of CPT are maintained when adopting different total number of cyclic precision schedule cycles during training, i.e., CPT is not sensitive to $N$. A visualization example for the precision schedule can be found in Appendix A. Additionally, we find that CPT is generally effective when using different dynamic precision schedule patterns (i.e., not necessarily the cosine schedule in Eq. (1)). We implement CPT following Eq. (1) in this work and discuss the potential variants in Sec. 4.3.

We visualize the training curve of CPT on ResNet-74 with CIFAR-100 in Fig. 1 and find that it achieves a 0.91% higher accuracy paired with a 36.7% reduction in the required training BitOPs (bit operations), as compared to its **static** fixed precision counterpart. In addition, Fig. 2 (c) visualizes the corresponding loss landscape, showing the effectiveness of CPT, i.e., such a simple and automated precision schedule leads to a better convergence with lower sharpness.

### 3.3 CPT: PRECISION RANGE TEST

The concept of CPT is simple enough to be plugged into any model or task to boost the training efficiency. One remaining question is how to determine the precision bounds, i.e., $B_{min}^i$ and $B_{max}^i$ in Eq. (1), which we find can be automatically decided in the first cycle (i.e., $T_i = T_0$) of the precision schedule using a simple PRT at a negligible computational cost. Specifically, PRT starts from the lowest possible precision, e.g., 2-bit, and gradually increases the precision while monitoring the difference in the training accuracy magnitude averaged over several consecutive iterations; once this training accuracy difference is larger than a preset threshold, indicating that the training can at least partially converge, PRT would claim that the lower bound is identified. While the upper bound can be similarly determined, there exists an alternative which suggests simply adopting the precision of CPT's static precision counterpart. The remaining cycles use the same precision bounds.

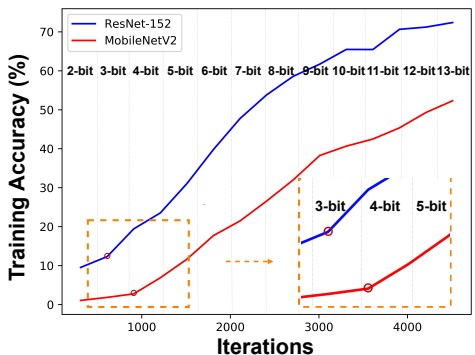

Figure 4: Illustrating the precision range test for ResNet-152 and MobileNetV2 on CIFAR-100, where the switching point which exceeds the preset threshold is denoted by red circles.

Fig. 4 visualizes the PRT for ResNet-152/MobileNetV2 trained on CIFAR-100. We can see that the lower precision bound identified when the model experiences a notable training accuracy improvement for ResNet-152 is 3-bit while that for MobileNetV2 is 4-bit, aligning with the common observation that ResNet-152 is more robust to quantization than the more compact model MobileNetV2.

## 4 EXPERIMENT RESULTS

In this section, we will first describe the experiment setup in Sec. 4.1, benchmarking results over SOTA training methods across various tasks in Sec. 4.2, and then comprehensive ablation studies of CPT in Sec. 4.3.

### 4.1 EXPERIMENT SETUP

**Models, datasets and baselines.** We consider underline{eleven models} (including eight ResNet based models (He et al., 2016), MobileNetV2 (Sandler et al., 2018), Transformer (Vaswani et al., 2017), and LSTM (Hochreiter & Schmidhuber, 1997)) and underline{five tasks} (including CIFAR-10/100 (Krizhevsky et al., 2009), ImageNet (Deng et al., 2009), WikiText-103 (Merity et al., 2016), and Penn Treebank (PTB) (Marcus et al., 1993)). Specifically, we follow (Wang et al., 2019b) for implementing MobileNetV2 on CIFAR-10/100. underline{Baselines:} We first benchmark CPT over three SOTA static low-precision training techniques: SBM (Banner et al., 2018), DoReFa (Zhou et al., 2016), and WAGEUBN (Yang et al., 2020), each of which adopts a different quantizer. Since SBM is the most competitive baseline among the three based on both their reported and our experiment results, we

apply CPT on top of SBM, and all the static precision baselines adopt the SBM quantizer unless specifically stated. Another baseline is the cyclic learning rate (CLR) (Loshchilov & Hutter, 2016) on top of static precision training, and we follow the best setting in (Loshchilov & Hutter, 2016).

**Training settings.** We follow the standard training setting in all the experiments. In particular, for classification tasks, we follow SOTA settings in (Wang et al., 2018b) for CIFAR-10/100 and (He et al., 2016) for ImageNet experiments, respectively; and for language modeling tasks, we follow (Baevski & Auli, 2018) for Transformer on WikiText-103 and (Merity et al., 2017) for LSTM on PTB.

**Precision settings.** The lower precision bounds in all the experiments are set using the PRT in Sec.3.3 and the upper bound is the same as the precision of the corresponding static precision baselines. We only apply CPT to the weights and activations (together annotated as FW) and use static precision for the errors and gradients (together annotated as BW), the latter of which is to ensure the stability of the gradients (Wang et al., 2018a) (more discussion in Appendix C). In particular, CPT from 3-bit to 8-bit with 8-bit gradient is annotated as FW(3,8)/BW8. The total number of periodic precision cycles, i.e., $N$ in Sec.3.3, for all the experiments is fixed to be 32 (see the ablation studies in Sec. 4.3).

**Hardware settings and metrics.** To validate the real hardware efficiency of the proposed CPT, we adopt standard FPGA implementation flows. Specifically, we employ the Vivado HLx design flow to implement FPGA-based accelerators on a Xilinx development board called ZC706 (Xilinx). To better evaluate the training cost, we consider both calculated GBitOPs (Giga bit operations) and real-measured latency on the ZC706 FPGA board.

### 4.2 BENCHMARK WITH SOTA STATIC PRECISION TRAINING METHODS

**Benchmark on CIFAR-10/100.** Benchmark over SOTA quantizers: We benchmark CPT with three SOTA static low-precision training methods, as summarized in Tab. 2, when training ResNet-74/164 and MobileNetV2 on CIFAR-10/100, covering both deep and compact DNNs which are representative difficult cases of low-precision DNN training. Note that **the accuracy improvement is the difference between CPT and the strongest baseline under the same setting**. Tab. 2 shows that (1) our CPT consistently achieves a win-win with both a higher accuracy (+0.19% ~ +1.25%) and a lower training cost (-21.0% ~ -37.1% computational cost and -14.7% ~ -21.4% latency) under all the cases on CIFAR-10/100, even for the compact model MobileNetV2, and (2) CPT notably outperforms the baselines in terms of accuracy **under extremely low precision**, which is one of the most useful scenarios for low-precision DNN training. In particular, CPT with periodic precision between 4-bit and 6-bit boosts the accuracy by +1.25% and +1.07% on ResNet-74/164, respectively, as compared to the static 6-bit training on CIFAR-10, verifying that CPT leads to a better convergence.

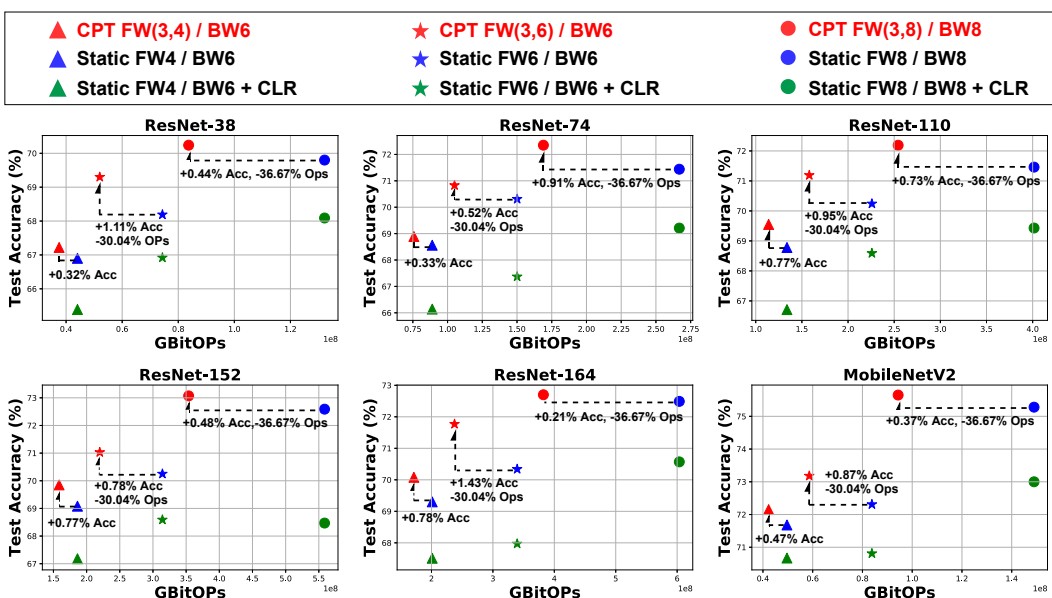

Figure 5: Test accuracy vs. the required GBitOPs when training ResNet-38/74/110/152/164 and MobileNetV2 on CIFAR-100 using static precision, static precision plus CLR, and CPT methods.

Table 2: The test accuracy, computational cost, and latency of CPT, DoReFa (Zhou et al., 2016), WAGEUBN (Yang et al., 2020), and SBM (Banner et al., 2018) for training the ResNet-74/164 and MobileNetV2 models on CIFAR-10/100.

| Model | Method | Precision (FW/BW) | CIFAR-10 Acc (%) | CIFAR-100 Acc (%) | GBitOPs | Latency (hour) |
|---|---|---|---|---|---|---|
| ResNet-74 | DoReFa | 8 / 8 | 91.16 | 69.31 | 2.67e8 | 44.6 |
| | WAGEUBN | 8 / 8 | 91.35 | 69.61 | 2.67e8 | 44.6 |
| | SBM | 8 / 8 | 92.57 | 71.44 | 2.67e8 | 44.6 |
| | **Proposed CPT** | **3 - 8 / 8** | **93.23** | **72.35** | **1.68e8** | **35.04** |
| | **Improv.** | | **+0.66** | **+0.91** | **-37.1%** | **-21.4%** |
| ResNet-74 | DoReFa | 6 / 6 | 90.94 | 69.01 | 1.50e8 | 33.2 |
| | WAGEUBN | 6 / 6 | 91.01 | 69.37 | 1.50e8 | 33.2 |
| | SBM | 6 / 6 | 91.15 | 70.31 | 1.50e8 | 33.2 |
| | **Proposed CPT** | **3 - 6 / 6** | **92.4** | **70.83** | **1.05e8** | **27.5** |
| | **Improv.** | | **+1.25** | **+0.52** | **-30.0%** | **-17.2%** |
| ResNet-164 | DoReFa | 8 / 8 | 91.40 | 70.90 | 6.04e8 | 101.9 |
| | WAGEUBN | 8 / 8 | 92.5 | 71.86 | 6.04e8 | 101.9 |
| | SBM | 8 / 8 | 93.63 | 72.53 | 6.04e8 | 101.9 |
| | **Proposed CPT** | **3 - 8 / 8** | **93.83** | **72.9** | **3.8e8** | **80.5** |
| | **Improv.** | | **+0.20** | **+0.37** | **-37.1%** | **-21.0%** |
| ResNet-164 | DoReFa | 6 / 6 | 91.13 | 70.53 | 3.40e8 | 76.7 |
| | WAGEUBN | 6 / 6 | 92.44 | 71.50 | 3.40e8 | 76.7 |
| | SBM | 6 / 6 | 91.95 | 70.34 | 3.40e8 | 76.7 |
| | **Proposed CPT** | **3 - 6 / 6** | **93.02** | **71.79** | **2.37e8** | **63.5** |
| | **Improv.** | | **+1.07** | **+0.29** | **-30.3%** | **-17.2%** |
| MobileNetV2 | DoReFa | 8 / 8 | 91.03 | 70.17 | 1.49e8 | 26.2 |
| | WAGEUBN | 8 / 8 | 92.32 | 71.45 | 1.49e8 | 26.2 |
| | SBM | 8 / 8 | 93.57 | 75.28 | 1.49e8 | 26.2 |
| | **Proposed CPT** | **4 - 8 / 8** | **93.76** | **75.65** | **1.04e8** | **21.6** |
| | **Improv.** | | **+0.19** | **+0.37** | **-30.2%** | **-17.6%** |
| MobileNetV2 | DoReFa | 6 / 6 | 90.25 | 68.4 | 8.39e7 | 18.4 |
| | WAGEUBN | 6 / 6 | 91.00 | 71.05 | 8.39e7 | 18.4 |
| | SBM | 6 / 6 | 91.56 | 72.31 | 8.39e7 | 18.4 |
| | **Proposed CPT** | **4 - 6 / 6** | **91.81** | **73.18** | **6.63e7** | **15.7** |
| | **Improv.** | | **+0.25** | **+0.87** | **-21.0%** | **-14.7%** |

**Benchmark over CLR on top of SBM:** We further benchmark CPT with CLR (Loshchilov & Hutter, 2016) (inherit its besting setting on CIFAR-100), with both being applied on top of SBM as it achieves the best performance among SOTA quantizers as shown in Tab. 2, based on more DNN models and precision as shown in Fig. 5. We can see that (1) CPT still consistently outperforms all the baselines with a better accuracy and efficiency trade-off, and (2) CLR on top of SBM leads to a negative effect on the test accuracy, which we conjecture is caused by both the instability of gradients and the sensitivity of gradients to the learning rate under low-precision training, as discussed in (Wang et al., 2018a), showing that CPT is more applicable to low-precision training than CLR.

**Loss landscape visualization:** To better understand the superior performance achieved by CPT, we visualize the loss landscape following the method in (Li et al., 2018), as shown in Fig. 6 covering both non-compact and compact models and two low-precision settings (i.e., 6-bit and 8-bit) which are bottlenecks for low-precision DNN training. We can observe that, again, both standard and compact DNNs trained with CPT experience wider contours with less sharpness, indicating that CPT helps DNN training optimization to converge to better local optima.

**Benchmark on ImageNet.** To verify the scalability of CPT on more complex tasks and larger models, we benchmark CPT with the SOTA static precision training method SBM (Banner et al., 2018) on ResNet-18/34/50 with ImageNet, under which it is challenging for low-precision training such as 4-bit to work. As shown in Tab. 3, we can observe that CPT still achieves a reduced computational cost (up to -30.4%) with a comparable accuracy (-0.20% ~ +0.06%). In particular, CPT works well

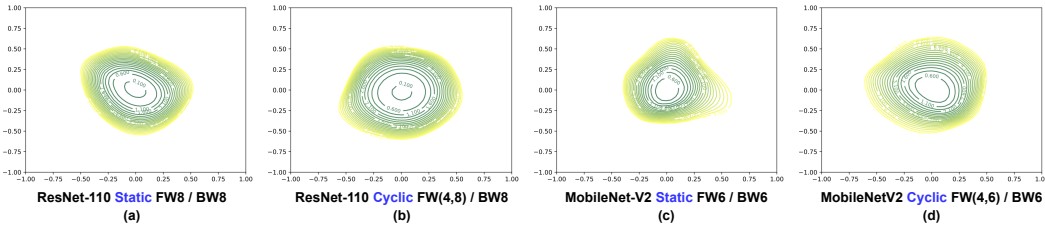

Figure 6: Loss landscape visualization of ResNet-110 and MobileNetV2 trained on CIFAR-100.

Table 3: The test accuracy and computational cost of ResNet-18/34/50 on ImageNet, trained with the proposed CPT and SBM (Banner et al., 2018).

| Model | Method | Precision (FW/BW) | Acc (%) | GBitOPs | Precision | Acc (%) | GBitOPs |
|---|---|---|---|---|---|---|---|
| ResNet-18 | SBM | 8 / 8 | 69.60 | 2.86e9 | 6 / 6 | 69.30 | 1.61e9 |
| | Proposed CPT | 4 - 8 / 8 | **69.64** | **1.99e9** | 4 - 6 / 6 | **69.33** | **1.27e9** |
| | **Improv.** | | **+0.04** | **-30.4%** | | **+0.03** | **-21.1%** |
| ResNet-34 | SBM | 8 / 8 | **73.32** | 5.77e9 | 6 / 6 | **72.82** | 3.24e9 |
| | Proposed CPT | 4 - 8 / 8 | 73.12 | **4.03e7** | 4 - 6 / 6 | 72.74 | **2.57e9** |
| | **Improv.** | | -0.20 | **-30.2%** | | -0.08 | **-20.7%** |
| ResNet-50 | SBM | 8 / 8 | 76.29 | 6.47e9 | 6 / 6 | 75.72 | 3.63e9 |
| | Proposed CPT | 4 - 8 / 8 | **76.35** | **4.51e9** | 4 - 6 / 6 | **75.74** | **2.87e9** |
| | **Improv.** | | **+0.06** | **-30.3%** | | **+0.02** | **-20.9%** |

on ResNet-50, leading to both a slightly higher accuracy and a better training efficiency, indicating the scalability of CPT with model complexity, and thus, its potential application in large scale training, in addition to on-device training scenarios.

**CPT for boosting accuracy:** An important perspective of CPT is its potential to improve training optimality in addition to efficiency. We illustrate CPT's advantage in improving the final accuracy through training ResNet-18/34 on ImageNet using CPT and static full precision. As shown in Tab. 4, CPT on ResNet-18/34 achieves a 0.91%/0.84% higher accuracy than their full precision counterparts on ImageNet, indicating that CPT can be adopted as a general technique to improve the final accuracy in addition to efficient training.

Table 4: The test accuracy of ResNet-18/34 on ImageNet: CPT (8-32) vs. full precision.

| Network | Method | Precision | Acc (%) |
|---|---|---|---|
| ResNet-18 | Full Precision | 32 | 69.76 |
| | **Proposed CPT** | 8-32 | **70.67** |
| | **Improv.** | | **+0.91** |
| ResNet-34 | Full Precision | 32 | 73.30 |
| | **Proposed CPT** | 8-32 | **74.14** |
| | **Improv.** | | **+0.84** |

**Benchmark on WikiText-103 and PTB.** We also apply CPT on language modeling tasks (including WikiText-103 and PTB) (see Tab. 5) to show that CPT is also applicable to natural language processing models. Tab. 5 shows that (1) CPT again consistently achieves a win-win in terms of accuracy (i.e., perplexity - the lower the better) and training efficiency, and (2) language modeling models/tasks are more sensitive to quantization, especially in LSTM models, as it always adapts to a larger lower precision bound, which is consistent with SOTA observations (Hou et al., 2019).

Table 5: The test accuracy and computational cost of (1) Transformer on WikiText-103 and (2) 2-LSTM (two-layer LSTM) on PTB, trained with CPT and SBM (Banner et al., 2018).

| Model / Dataset | Method | Precision (FW/BW) | Perplexity | GBitOPs | Precision | Perplexity | GBitOPs |
|---|---|---|---|---|---|---|---|
| Transformer WikiText-103 | SBM | 8 / 8 | 31.77 | 1.44e6 | 6 / 8 | 32.41 | 9.87e5 |
| | **Proposed CPT** | **4 - 8 / 8** | **30.22** | **1.0e6** | **4 - 6 / 8** | **31.0** | **7.66e5** |
| | **Improv.** | | **-1.55** | **-30.2%** | | **-1.41** | **-22.4%** |
| 2-LSTM PTB | SBM | 8 / 8 | 96.95 | 4.03e3 | 6 / 8 | 97.47 | 2.77e3 |
| | **Proposed CPT** | **5 - 8 / 8** | **96.39** | **3.09e3** | **5 - 6 / 8** | **97.0** | **2.48e3** |
| | **Improv.** | | **-0.56** | **-23.2%** | | **-0.47** | **-10.5%** |

### 4.3 ABLATION STUDIES OF CPT

**CPT with different precision ranges.** We also evaluate CPT under a wide range of upper precision bounds, which correspond to a different target efficiency, to see if CPT still works well. Fig. 7 plots a boxplot with each experiment being repeated ten times. We can see that (1) regardless of the adopted precision ranges, CPT consistently achieves a win-win (a +0.74% ~ +2.03% higher accuracy and a -18.3% ~ -48.0% reduction in computational cost), especially under lower precision scenarios, and (2) **CPT even shrinks the accuracy variance**, which better aligns with the practical goal of efficient training.

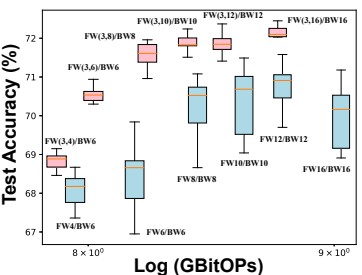

Figure 7: Training ResNet-74 on CIFAR-100 with CPT and its static counterpart.

**CPT with different adopted numbers of precision schedule cycles.** To evaluate CPT's sensitivity to the total number of adopted cyclic precision schedule cycles, we apply CPT on ResNet-38/74 and CIFAR-100 when using different numbers of schedule cycles. Fig. 8 plots the mean of the achieved

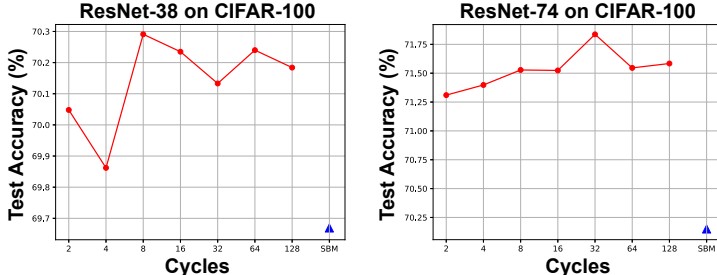

Figure 8: The achieved test accuracy of CPT under different adopted numbers of precision schedule cycles, as compared to the static precision baseline SBM, when training ResNet-38/74 on CIFAR-100.

accuracy based on ten repeated experiments. We can see that (1) different choices for the total number of cycles lead to a comparable accuracy (within 0.5% accuracy) on CIFAR-10/100; and (2) CPT with different number of precision schedule cycles consistently outperforms the static baseline SBM in the achieved accuracy. Based on this experiment, we set $N = 32$ for simplicity.

**CPT with different cyclic precision schedule patterns.** We also evaluate CPT using other cyclic precision schedule patterns in addition to the cosine one, including a triangular schedule motivated by (Smith, 2017) and a cosine annealing schedule as the learning rate schedule in (Loshchilov & Hutter, 2016), with all adopting 32 cyclic cycles for fairness. Experiments in Tab. 6 show that (1) CPT with different schedule patterns is consistently effective, and (2) CPT with the other two schedule patterns even surpasses the cosine one in some cases but underperforms on compact models. We leave how to determine the optimal cyclic pattern for a given model and task as a future work.

Table 6: CPT with different precision schedules into cyclic precision training for ResNet-74/164 and MobileNetV2 on CIFAR-100. Cosine (CPT) is the current schedule adopted by CPT.

| | ResNet-74 | | ResNet-164 | | MobileNetV2 | |
|---|---|---|---|---|---|---|
| Schedule | FW(3,8) / BW8 | FW(3,6) / BW6 | FW(3,8) / BW8 | FW(3,6) / BW6 | FW(4,8) / BW8 | FW(4,6) / BW6 |
| Cosine (our CPT) | **72.35** | 70.83 | 72.7 | 71.77 | **75.65** | **73.18** |
| Triangular | 71.61 | **71.22** | **72.94** | **72.37** | 74.92 | 71.37 |
| Cosine anneal | 71.37 | 70.80 | 72.4 | 72.03 | 74.59 | 73.12 |

## 5  DISCUSSIONS ABOUT FUTURE WORK

**The theoretical perspective of CPT.** There has been a growing interest in understanding and optimizing DNN training. For example, (Li et al., 2019) shows that training DNNs with a large initial learning rate helps the model to memorize more generalizable patterns faster and better. Recently, (Zhu et al., 2020) showed that under a convexity assumption the convergence bound of reduced-precision DNN training is determined by a linear combination of the quantization noise and learning rate. These findings regarding DNN training seem to be consistent with the effectiveness of our CPT.

**The hardware support for CPT.** Recent progresses in mixed-precision DNN accelerators (Lee et al., 2019; Kim et al., 2020) with dedicated modules for supporting dynamic precisions are promising to support our CPT. We leave the optimal accelerator design for CPT as our future works.

## 6  CONCLUSION

We hypothesize that DNNs' precision has a similar effect as the learning rate when it comes to DNN training, i.e., low precision with large quantization noise helps DNN training exploration while high precision with more accurate updates aids model convergence, and thus advocate that dynamic precision schedules help DNN training optimization. We then propose the CPT framework which adopts a periodic precision schedule for low-precision DNN training in order to boost the achievable Pareto frontier of task accuracy and training efficiency. Extensive experiments and ablation studies verify that CPT can reduce computational cost during training while achieving a comparable or even better accuracy. Our future work will strive to identify more theoretical grounds for such dynamic low-precision training.

## ACKNOWLEDGEMENT

The work is supported by the National Science Foundation (NSF) through the Real-Time Machine Learning (RTML) program (Award number: 1937592) and the NSF Award 1801865.

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

## A VISUALIZATION OF THE PRECISION SCHEDULE IN CPT

Fig. 9 visualizes the precision schedule FW(3,8) with eight cycles for training on CIFAR-10/100.

Table 7: The test accuracy of ResNet-38/74 on CIFAR-10 trained with CPT enabled at different epochs. Note that the lowest training precision is applied before enabling CPT in all the experiments.

| Network | Starting Epoch | 0 | 60 | 80 | 100 | 120 | 160 (No CPT) |
|---------|---------------|-----|-----|-----|-----|-----|--------------|
| ResNet-38 | FW(3,8) / BW8 | **70.24** | 69.60 | 68.37 | 68.04 | 67.65 | 62.3 |
| | FW(3,6) / BW6 | **69.30** | 68.65 | 68.00 | 67.83 | 67.85 | 62.3 |
| ResNet-74 | FW(3,8) / BW8 | **72.35** | 70.55 | 69.44 | 69.19 | 68.96 | 64.2 |
| | FW(3,6) / BW6 | **70.83** | 70.32 | 69.25 | 68.72 | 68.56 | 64.2 |

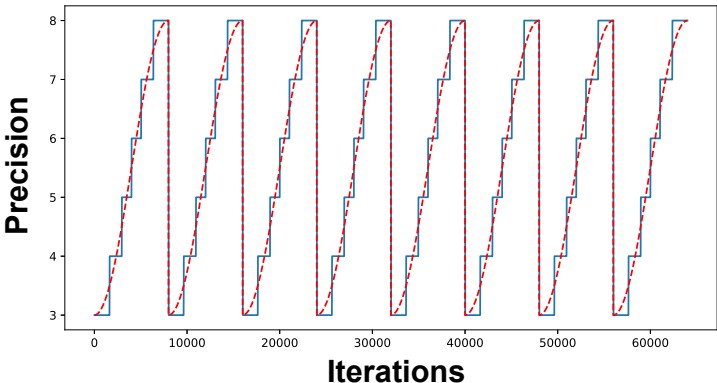

Figure 9: Visualization of the precision schedule FW(3,8) with eight cycles, where the red line is the cosine-manner schedule and the blue line is the adopted precision after rounding.

## B  ABLATION STUDY: THE STARTING EPOCH FOR ENABLING CPT

We conduct another ablation study to explore whether CPT is always necessary during the whole training process. We start from training with the lowest precision in the precision range of CPT, and then enable CPT at different epochs to see the influence on the final accuracy. Specifically, we train ResNet-38/74 on CIFAR-100 for 160 epochs, considering two CPT precision ranges with different starting epochs to enable CPT as shown in Tab. 7. We consistently observe that (1) an early starting epoch of CPT leads to a better accuracy on all the considered precision ranges and networks, and (2) even enabling CPT at a later training stage still leads to a notable better accuracy than training without CPT.

## C  ABLATION STUDY: CPT ON TOP OF GRADIENTS

We decide not to apply CPT on top of gradients since (1) the resulting instability of low precision gradients during training (Wang et al., 2018a) can harm the final convergence, and (2) generally the required precision of gradients is higher than that of weights and activations, so that the benefit of applying CPT on top of gradients in terms of efficiency is limited.

To validate this, we show the results of applying CPT on top of gradients with fixed precision for weights and activations in Tab. 8. As expected, CPT on top of gradients can hardly benefit either accuracy or efficiency as compared with its static precision baseline. Therefore, we decide to adopt fixed precision for gradients in all other experiments.

Table 8: Training ResNet-74 on CIFAR-100 with static precision and CPT on top of gradient only.

| FW | BW | Accuracy (%) |
|----|----|----|
| 6 | 6 | 70.31 |
| 6 | 8 | 70.51 |
| 6 | 6-8 | 69.45 |
| 6 | 6-10 | 70.32 |
| 6 | 6-12 | 69.59 |
| 6 | 6-16 | 70.24 |

