# OpenReview forum: "CPT: Efficient Deep Neural Network Training via Cyclic Precision"
_ICLR.cc/2021/Conference — ICLR 2021 Spotlight_

### Official Review · AnonReviewer4 · 2020-10-28
**technique to cyclically vary precision during training**

**Rating:** 7
**Confidence:** 3

**Review:**

A simple yet apparently effective technique. The paper is clear and well written.

The authors demonstrate that cyclically changing the precision of weights and activations during training leads to better results (both accuracy and training cost) than static quantisation methods. The method is simple and easy to implement . The  authors provide a method to automatically select the method's hyperparameters.

1. This work appears to be an another example of multiprecision training (with a precision-switching mechanism): https://arxiv.org/abs/2006.09049

2. The abstract claims that the variance of accuracy decreases when using CPT. This is briefly mentioned around Figure 7, but I would also expect to see these figures available for comparison in Tables 1-3, and some more discussion.

3. The original hypothesis was that cyclic precision would have a similar effect to cyclic learning rate, but there's little discussion of this in the paper. There's one paragraph in Section 4.2 which compares CPT against CLR.  Could both techniques be applied together?

Minor

* Table 1: Apologies if I am missing something obvious, it's unclear to me what "GBitOPs" and "latency" refer to. It appears that two separate models have been trained, one for CIFAR-10 and one for CIFAR-100, but only one GBitOPs/latency figure is reported?

* I find it quite hard to parse paragraph 2 of section 4.2 (Benchmark over CLR on top of SBM....)

* Section 4.2, paragraph 4, "staic" should be "static"

* There are several capitalisation issues in the references

EDIT: Thanks to the authors' for their clear responses, I'm happy to raise my score to a 7.

---

> ### Author Response · Authors · 2020-11-22
> **Response to Reviewer 4: Part 1**
>
> Thanks for your valuable suggestions and below are our answers to your questions:
>
> (Note that here's the Part 1 of our responses, see the Part 2 for more responses.)
>
> **1. Comparison with another multiprecision training work.**
>
> Thanks for recommending the reference which can be viewed as a concurrent work with ours. While both share a similar goal of efficient training, CPT is different in two aspects: (1) In terms of methodology, their method needs to calculate the metric named gradient diversity on the fly during training which introduces additional overhead and relies on many hyper-parameters, whereas our CPT is fully automatic with little computation overhead or hyper-parameter tuning; and (2) In terms of the achieved performance, their precision choices are 8-, 12-, 14-, and 16-bit, and they achieve an accuracy of 69.09% when training ResNet-18 on ImageNet. In comparison, our CPT with a precision range of (4-bit, 8-bit) (see table 2 in our paper) achieves an accuracy of 69.33%, i.e., a 0.24% better accuracy with a notably less training cost. Furthermore, we conduct additional experiments and find that CPT training of ResNet-18 with a precision range of (8-bit, 16-bit) achieves an accuracy of 70.67%, which is even 0.91% higher than that of training with full precision, showing the superiority of our CPT over the baseline in terms of improving DNN training optimality.
>
> **2. Variance of table 1-3.**
>
> Thanks for your suggestion. Given the limited time, we have repeated the experiments 10 times in tables 1 and 3 to find that CPT still consistently achieves a lower or comparable variance (e.g., 0.09 / 0.10  of CPT vs. 0.25 / 0.15 of the static training for training ResNet-164 / MobileNet-V2 on CIFAR-100) together with a better average performance, as compared to static precision training. We will add more results to the final version.
>
> **3. More justifications about the relationship between precision and learning rate.**
>
> Thanks for pointing out and we agree that more empirical and theoretical justifications will help improve our work. To further verify the hypothesis empirically, we conduct a set of experiments to see whether a low precision can have a similar effect as a high learning rate in DNN training.
>
> Settings: We train ResNet-38 / 74 on CIFAR-100 for 160 epochs using an SGD optimizer and a momentum of 0.9 with a piecewise learning rate decay following the basic training setting mentioned in Sec. 4.1 of our manuscript. In particular, we divide the training of 160 epochs into three stages: [0-th, 80-th], [80-th,120-th], and [120-th, 160-th]; for the stage of [0-th, 80-th], we adopt different learning rates and precisions for the weights and activations, while using full precision for the remaining two stages with a learning rate of 0.01 for the [80-th,120-th] epochs and 0.001 for the [120-th, 160-th] epochs. The final accuracy of different settings is reported below:
>
> ***ResNet-74 on CIFAR-100:***
>
> | lr([0-th, 80-th]) | 0.1 | 0.06 | 0.03 | 0.01 |
> |:-:|:-:|:-:|:-:|:-:|
> | 4-bit([0-th, 80-th]) | 70.96 | 69.54 | 68.26 | 67.19 |
> | 6-bit([0-th, 80-th]) | 71.62 | 70.28 | 68.84 | 66.16 |
> | 8-bit([0-th, 80-th]) | 71.60 | 70.67 | 68.45 | 65.85 |
> | FP([0-th, 160-th]) | 71.66 | 70.00 | 68.69 | 65.62 |
>
> ***ResNet-38 on CIFAR-100:***
>
> | lr([0-th, 80-th]) | 0.1 | 0.06 | 0.03 | 0.01 |
> |:-:|:-:|:-:|:-:|:-:|
> | 4-bit([0-th, 80-th]) | 69.45 | 68.63 | 67.69 | 65.90 |
> | 6-bit([0-th, 80-th]) | 70.22 | 68.87 | 67.15 | 66.10 |
> | 8-bit([0-th, 80-th]) | 69.96 | 68.66 | 66.75 | 64.99 |
> | FP([0-th, 160-th]) | 70.45 | 69.53 | 67.47 | 64.50 |
>
> Analysis: We can observe that as the learning rate sufficiently reduces for the first stage of [0-th, 80-th], adopting a lower precision for this stage will lead to a higher accuracy than training with full precision. In particular, with the standard initial learning rate 0.1, full precision training achieves a 1.00% / 0.70% higher accuracy than the 4-bit one on ResNet-38 / 74, respectively; whereas as the initial learning rate decreases, this accuracy gap gradually narrows and then reverses, e.g., when the initial learning rate becomes 1e-2, training with [0-th, 80-th] of 4-bit achieves a 1.40% / 1.57% higher accuracy than the full precision ones.
>
> This set of experiments show that (1) when the initial learning rate is low, training with lower initial precisions consistently lead to a better accuracy than training with full precision, indicating that lowering the precision introduces a similar effect  of favoring exploration as that of a high learning rate; and (2) although a low precision can alleviate the accuracy drop caused by a low learning rate, a high learning rate is in general necessary to maximize the accuracy.

---

> ### Author Response · Authors · 2020-11-22
> **Response to Reviewer 4: Part 2**
>
> (Note that here's the Part 2 of our responses, see the Part 1 for more responses.)
>
> **4. CPT on top of CLR.**
>
> We conduct additional experiments to benchmark over the baseline of adopting CPT on top of CLR on CIFAR-100. As shown in the table below, the observations are consistent with the conclusion in Sec. 4.2, i.e., CLR will have a negative impact under low-precision training as compared with the case of CPT only, due to the instability of gradients and the sensitivity of gradients to learning rate under low-precision training as analyzed in ‘Training deep neural networks with 8-bit floating point numbers’ (N. Wang, NeurIPS’18).
>
> |  | FW(3,8)/BW8 | FW(3,8)/BW8 | FW(3,6)/BW6 | FW(3,6)/BW6 | FW(3,4)/BW6 | FW(3,4)/BW6 |
> |:-:|:-:|:-:|:-:|:-:|:-:|:-:|
> |  | CLR+CPT | CPT only | CLR+CPT | CPT only | CLR+CPT | CPT only |
> | ResNet-38 | 69.67 | **70.24** | 69.11 | **69.30** | 67.06 | **67.22** |
> | ResNet-74 | 71.06 | **71.43** | 69.83 | **70.83** | 68.69 | **68.89** |
> | ResNet-164 | 72.19 | **72.90** | 70.97 | **71.79** | 69.24 | **70.08** |
>
> **5. Clarify more details.**
>
> In table 1, “GBitOPs” means Giga Bit OPerations and “latency” is the total training time. Since CIFAR-10 and CIFAR-100 have the same number of images with the same resolution, and the training settings (e.g., epochs) are the same, the training costs of the same network on the two datasets are the same. Thanks very much for pointing out the typos and presentation issues, and we will revise and carefully proofread in the final version.

---

### Official Review · AnonReviewer1 · 2020-10-29
**cyclic precision is very interesting but the paper requires more explanations and motivations behind the idea**

**Rating:** 7
**Confidence:** 3

**Review:**

The authors proposed an interesting low-precision training method using a dynamic precision schedule. Their proposed Cyclic Precision Training (CPT) cyclically varies the precision during the training and the boundary of precision values is determined by a precision range test (PRT). As shown in their empirical results, CPT largely reduces time/energy costs during training while maintaining comparable accuracy. While CPT does show a promising empirical performance, the motivation and reason behind the proposed method require further explanations.
Specifically,

1. Motivation hypothesis in section 3.1 lacks explanation and justification. Authors point out that dynamic learning rate schedules help to improve the training and DNNs learn to fit different patterns at different training stages, but there is no strong connection to why dynamic learning rate could relate to dynamic precision schedule.

2. Visualization examples aiming for supporting the hypothesis are not strong enough. The progressive precision case only achieves slightly higher accuracy and such benefit may be diminished given a large number of repeated experiments. Therefore I don't fully agree authors' statement "progressive precision schedule helps to converge to better local minima with wider contours, indicating a lower generalization error".

3. Authors do point out that low-precision with large quantization noise may have a similar effect as a high learning rate with large stochastic noise, which helps the exploration over loss surface during DNN training. This hypothesis is interesting and it would be better to support the hypothesis with empirical validation and theoretical justification.

In addition, I think the proposed precision range test (PRT) method is unfair and requires modification.  PRT starts from the lowest possible precision and gradually increases the precision while monitoring the difference in the training accuracy magnitude averaged over several consecutive iterations. However, such a difference could also be affected by the training stages at different iterations. A better test should be evaluating all precision settings from the same starting point (e.g. training from scratch), and then measuring the differences after reaching the convergence stage.

The paper is mostly well-written. Section 4.4 seems redundant and should be put before the experimental section. The connections are also required to be explained in detail.

---

> ### Author Response · Authors · 2020-11-22
> **Response to Reviewer 1: Part 1**
>
> Thanks for your valuable suggestions and below are our answers to these questions:
>
> (Note that here's the Part 1 of our responses, see the Part 2 for more responses.)
>
> **1. Justification about the hypothesis of the similar role of precision and learning rate.**
>
> Thanks for pointing out and we agree that more empirical and theoretical justifications will help improve our work. To further verify the hypothesis empirically, we conduct a set of experiments to see whether a low precision can have a similar effect as a high learning rate in DNN training.
>
> Settings: We train ResNet-38 / 74 on CIFAR-100 for 160 epochs using an SGD optimizer and a momentum of 0.9 with a piecewise learning rate decay following the basic training setting mentioned in Sec. 4.1 of our manuscript. In particular, we divide the training of 160 epochs into three stages: [0-th, 80-th], [80-th,120-th], and [120-th, 160-th]; for the stage of [0-th, 80-th], we adopt different learning rates and precisions for the weights and activations, while using full precision for the remaining two stages with a learning rate of 0.01 for the [80-th,120-th] epochs and 0.001 for the [120-th, 160-th] epochs. The final accuracy of different settings is reported below:
>
> ***ResNet-74 on CIFAR-100:***
>
> | lr([0-th, 80-th]) | 0.1 | 0.06 | 0.03 | 0.01 |
> |:-:|:-:|:-:|:-:|:-:|
> | 4-bit([0-th, 80-th]) | 70.96 | 69.54 | 68.26 | 67.19 |
> | 6-bit([0-th, 80-th]) | 71.62 | 70.28 | 68.84 | 66.16 |
> | 8-bit([0-th, 80-th]) | 71.60 | 70.67 | 68.45 | 65.85 |
> | FP([0-th, 160-th]) | 71.66 | 70.00 | 68.69 | 65.62 |
>
> ***ResNet-38 on CIFAR-100:***
>
> | lr([0-th, 80-th]) | 0.1 | 0.06 | 0.03 | 0.01 |
> |:-:|:-:|:-:|:-:|:-:|
> | 4-bit([0-th, 80-th]) | 69.45 | 68.63 | 67.69 | 65.90 |
> | 6-bit([0-th, 80-th]) | 70.22 | 68.87 | 67.15 | 66.10 |
> | 8-bit([0-th, 80-th]) | 69.96 | 68.66 | 66.75 | 64.99 |
> | FP([0-th, 160-th]) | 70.45 | 69.53 | 67.47 | 64.50 |
>
> Analysis: We can observe that as the learning rate sufficiently reduces for the first stage of [0-th, 80-th], adopting a lower precision for this stage will lead to a higher accuracy than training with full precision. In particular, with the standard initial learning rate 0.1, full precision training achieves a 1.00% / 0.70% higher accuracy than the 4-bit one on ResNet-38 / 74, respectively; whereas as the initial learning rate decreases, this accuracy gap gradually narrows and then reverses, e.g., when the initial learning rate becomes 1e-2, training with [0-th, 80-th] of 4-bit achieves a 1.40% / 1.57% higher accuracy than the full precision ones.
>
> This set of experiments show that (1) when the initial learning rate is low, training with lower initial precisions consistently lead to a better accuracy than training with full precision, indicating that lowering the precision introduces a similar effect  of favoring exploration as that of a high learning rate; and (2) although a low precision can alleviate the accuracy drop caused by a low learning rate, a high learning rate is in general necessary to maximize the accuracy.
>
> **2. Connections between dynamic learning rate and dynamic precision.**
>
> Thanks for pointing out and we will further clarify this in the final version. Based on our above justification in Question 1, we motivate dynamic precision from two aspects: (1) “Cyclical Learning Rates for Training Neural Networks” (L. Smith, WACV’17) points out that cyclic learning rate explores the optimal learning rate between the lower and upper bound to improve convergence. Given the similar role of learning rate and precision in terms of DNN training exploration, cyclic precision can potentially explore the optimal precision setting within the precision range; and (2) based on the experiments provided in response to Question 1 about the role of low precision which leads to better exploration with inaccurate updates, cyclic precision can help balance the exploration effect of low precision and the accurate update of high precision, resulting in better optima.

---

> ### Author Response · Authors · 2020-11-22
> **Response to Reviewer 1: Part 2**
>
> (Note that here's the Part 2 of our responses, see the Part 1 for more responses.)
>
> **3. More justifications for progressive precision schedules.**
>
> As we can see in the experiments provided in response to the above Question 1, progressive precision schedules which start with lower precision indeed help DNN training exploration and lead to better achieved accuracy. Furthermore, we conduct additional experiments on CIFAR-100 with various precision ranges following the setting in Sec. 3.1 as shown in the table below. We can observe that progressive precision schedules can consistently achieve less training cost with better or comparable accuracy, which further verifies our hypothesis.
>
> |  | ResNet-38 | ResNet-74 |
> |:-:|:-:|:-:|
> | FW6/BW6 | 68.19 | 70.31 |
> | FW(3,6)/BW6 | 69.02 (+0.83) | 70.54 (+0.23) |
> | FW8/BW8 | 69.81 | 71.44 |
> | FW(3,8)/BW8 | 70.08 (+0.27) | 71.62 (+0.18) |
> | FW12/BW12 | 70.06 | 71.24 |
> | FW(3,12)/BW12 | 70.16 (+0.10) | 71.69 (+0.45) |
>
> **4. Implementation of PRT.**
>
> Thanks for your suggestions about PRT. We agree that your proposed PRT may be more fair, while it needs to train each precision from scratch which violates our goal of efficient training. In contrast, our indicator for PRT can be naturally embedded into the first stage of CPT with little training overhead. Furthermore, our indicator has been evaluated to be effective across networks and datasets, and we consistently observe a notable switch point in the training accuracy when switching to the proper precision. Finally, since in most tasks the indicator only needs to make a decision between 3-bit or 4-bit which have been empirically found by existing low-precision training works to be the lowest training precisions that won’t lead to notable accuracy drops on different tasks, we humbly argue that our  simple indicator here is sufficient. We will further clarify this in the final version.
>
> **5. Rearrange Sec. 4.4.**
>
> Thanks for pointing out and we will rearrange Sec. 4.4 and add the above justifications to support the claims in the final version.

---

### Official Review · AnonReviewer2 · 2020-10-29
**Review for CPT: Efficient Deep Neural Network Training via Cyclic Precision**

**Rating:** 7
**Confidence:** 5

**Review:**

### Overview
In this paper, the authors proposed Cyclic Precision Training (CPT) for low precision training. CPT varies the precision between two boundary values cyclically during training. It improves the accuracy, converges to a wider minima, reduces training variance, and reduces the training cost.

### Clarity
The paper is well written and clearly organized. The figures and tables are informative and clearly organized.

### Pros
1. The paper provides an in-depth analysis of the advantages of varying precision. It shows the connection between precision and learning rate, and dynamic precision helps convergence to a better minima, verified by the visualization.
2. The proposed CPT is effective for improving training convergence and reducing training cost. As shown in Figure 5, the accuracy v.s. training cost trade-off is better compared to static low-precision training. CPT even slightly improves the final accuracy.
3. The author proposed an automatic method to determine the boundaries for cyclic precision at negligible overhead.
4. The experiments are solid. The advantage of CPT is consistent over different models and tasks. Ablation study shows the robustness of CPT w.r.t. different hyper-parameters.

### Cons
1. The results of CPT is impressive. However, the actual application scenario could be quite restricted at this time. Most of the hardware does not support flexible bit-precision like 3-8 in this paper. Supporting multiple bit-precision will require extra compute units, leading to higher cost and energy consumption. It is unclear if the trade-off is worthwhile. Using FPGA can support flexible bit like in this paper, but using ASIC supporting int8 training (like GPU) might be more efficient compared to the improvement from CPT (I think generally FPGA is not more efficient than GPU for NN training).
2. The precision range test is not explained clearly. Is the threshold consistent for all experiments? In figure 4, the training accuracy for the switching point on two curves are not the same. Does it indicate we need to manually tune the threshold for each experiment?
3. The authors claim that CPT improves convergence for quantization training. Since the hypothesis is that the precision may work similarly to the learning rate, a baseline would be training with fixed precision (just like in this paper) while using cyclic *learning rate*.  Can CPT also outperform this baseline?

EDIT: The response is very helpful and it addresses my concerns. I raised my score to 7 after reading the response.

---

> ### Author Response · Authors · 2020-11-22
> **Response to Reviewer 2**
>
> Thanks for your valuable suggestions and below are our answers to these questions:
>
> **1. Limited application scenario and hardware support of CPT and inferior training efficiency of FPGA than GPU.**
>
> First, recent progresses in mixed-precision DNN accelerators with dedicated modules for supporting dynamic precisions are promising to support our CPT, such as “LNPU: A 25.3TFLOPS/W Sparse Deep-Neural-Network Learning Processor with Fine-Grained Mixed Precision of FP8-FP16” (J. Lee, ISSCC’19) and “1b-16b variable bit precision DNN processor for emotional hri system in mobile devices” (C. Kim, JICS’20). We leave the optimal accelerator design for CPT as our future works.
>
> Second, FPGA has gradually gained more attention in DNN training thanks to its much improved efficiency, as mentioned in “FPGA-based Accelerators of Deep Learning Networks for Learning and Classification: A Review” (A. Shawahna, IEEE Access’18); and recent works have shown the superiority of FPGA clusters over GPU clusters in DNN training such as “Towards Efficient Deep Neural Network Training by FPGA-based Batch-level Parallelism” (C. Luo, FCCM’19) and “FPDeep: Scalable Acceleration of CNN Training on Deeply-Pipelined FPGA Clusters” (T. Geng, IEEE Trans Comput’20), indicating the future trend of training DNNs on FPGA. We will discuss more about the hardware support for CPT in the final version.
>
> Finally, we would like to also justify from the perspective of CPT's advantageous accuracy. In particular, in addition to the potential impact on efficient training, another important perspective and application scenario of our CPT is that it favors better exploration towards a better convergence, improving the achieved accuracy. Notably, CPT’s results (e.g., those in Table 1) even surpass its full precision counterparts. To further justify this point, we conduct additional experiments to train ResNet-18/34 on ImageNet with a precision range FW(8,16), and benchmark over their full precision training results. We find that CPT on ResNet-18 / 34 achieves 70.67% / 74.14% top-1 accuracy, i.e., even 0.91% / 0.84% higher than their full precision counterparts, respectively, indicating the superiority of CPT in improving training optimization which serves as another application scenario of our CPT.
>
> **2. Details about the precision range test.**
>
> Thanks for pointing out and we will further clarify in the final version. We adopt the same threshold for all the experiments of the same type of DNNs, i.e., we have three different thresholds for CNN, Transformer, and LSTM, respectively, considering their different convergence properties. For example, we calibrate the threshold on ResNet-74 trained on CIFAR-100 and apply it to all other CNNs on both CIFAR-100 and ImageNet. Here is the reason that our threshold is effective/insensitive across different networks of the same type: As mentioned in Sec 3.3, we adopt the training accuracy difference between two consecutive epochs normalized over the average training accuracy of all previous epochs as the indicator, instead of directly using the absolute values of  training accuracy. Our indicator is evaluated to be consistently effective, since in most tasks the indicator only needs to select between 3-bit or 4-bit which have been empirically found by existing low-precision training works to be the lowest training precisions that won’t lead to notable accuracy drops on different tasks.
>
> **3. Comparison with cyclic learning rate (CLR) on top of fixed precision training.**
>
> We have shown this baseline in Fig.5 as analysed in Sec. 4.2. We find that CLR on top of fixed precision training under low precision settings leads to a negative effect on the accuracy, which we conjecture is caused by both the instability of gradients and the sensitivity of gradients to learning rate under low-precision training, as discussed in ‘Training deep neural networks with 8-bit floating point numbers’ (N. Wang, NeurIPS’18), showing that CPT are more applicable to low-precision training than CLR although they have a similar effect in terms of exploration.
>
> Furthermore, we have conducted more experiments to support our hypothesis of the similar effect of learning rate and precision in DNN training which can be found in our answer to Review #3.

---

### Official Review · AnonReviewer3 · 2020-10-30
**Interesting empirical paper, unsure about theoretical explanation**

**Rating:** 7
**Confidence:** 5

**Review:**

The authors propose cyclic precision training (CPT), a method to train integer-quantized neural networks with high precision while saving bit operations. CPT alternates the numerical precision of the network during training between low (2-3 bits) and high (the final desired precision, e.g. 8 bits). A total of 32 cycles of low to high are used, and this is only applied to the weights and activations, the backward pass is always in high precision. CPT is able to achieve improved accuracy across a variety of models on CIFAR-10/100 as well as transformers on PTB and WikiText. On ImageNet, CPT achieves accuracy on part with regular quantized training but saves bit ops.

The authors include a large number of other interesting data to show that CPT is insensitive to hyperparameters. Regardless of what the low precision setting, CPT(a,b) achieves better accuracy than b-bit quantized training on CIFAR-100. Varying the number of cycles between 8 and 128  affects the results by at most 0.5%. These experiments and the loss visualizations show that CPT is robust on CIFAR.

The major thing holding the paper back for me is that the lack of a good theoretical explanation for the CPT phenomenon. The authors draw a parallel between cyclic learning rates and CPT, but the paper itself shows that cyclic LR is not effective with quantization. I believe that lower accuracy means higher noise which resembles a higher LR, but clearly the two are not equivalent. One other idea is that lower accuracy may resemble lower batch size (higher SGD noise), but not sure what that would mean.

In general, the paper presents overwhelming evidence that CPT is a real phenomenon which may help train quantized DNNs. Even though the mechanism is currently not well understood, the paper deserves to be seen by others.

Questions:
 - I assume the CNNs are all trained with SGD. Was the transformer trained with SGD or ADAM or another optimizer?
 - What if you just trained in low precision until say, epoch 80, then use CPT? Does cyclic precision have to be used throughout training or just near the end?

EDIT: I managed to set this up on a small transformer and was able to replicate the paper results. The transformer was trained with ADAM. Definite accept from me.

---

> ### Author Response · Authors · 2020-11-22
> **Response to Reviewer 3**
>
> Thanks for recognizing the potential impact of our work and for your valuable suggestions. Below are our detailed response:
>
> **1. Lack of a theoretical explanation for CPT.**
>
> Yes, more empirical and theoretical justifications in this regard will help improve our work. As mentioned in Sec. 3.1 of our manuscript, our hypothesis is that a low precision with large quantization noise helps better explore the solution space, which will benefit the final convergence, similar to the case of adopting a high learning rate even though they are not exactly equivalent as you mentioned. To further verify the hypothesis empirically, we conduct a set of experiments to see whether a low precision can serve a similar role as a high learning rate in DNN training.
>
> Settings: We train ResNet-38 / 74 on CIFAR-100 for 160 epochs using an SGD optimizer and a momentum of 0.9 with a piecewise learning rate decay following the basic training setting mentioned in Sec. 4.1 of our manuscript. In particular, we divide the training of 160 epochs into three stages: [0-th, 80-th], [80-th,120-th], and [120-th, 160-th]; for the stage of [0-th, 80-th], we adopt different learning rates and precisions for the weights and activations, while using full precision for the remaining two stages with a learning rate of 0.01 for the [80-th,120-th] epochs and 0.001 for the [120-th, 160-th] epochs. The final accuracy of different settings is reported below:
>
> ***ResNet-74 on CIFAR-100:***
>
> | lr([0-th, 80-th]) | 0.1 | 0.06 | 0.03 | 0.01 |
> |:-:|:-:|:-:|:-:|:-:|
> | 4-bit([0-th, 80-th]) | 70.96 | 69.54 | 68.26 | 67.19 |
> | 6-bit([0-th, 80-th]) | 71.62 | 70.28 | 68.84 | 66.16 |
> | 8-bit([0-th, 80-th]) | 71.60 | 70.67 | 68.45 | 65.85 |
> | FP([0-th, 160-th]) | 71.66 | 70.00 | 68.69 | 65.62 |
>
> ***ResNet-38 on CIFAR-100:***
>
> | lr([0-th, 80-th]) | 0.1 | 0.06 | 0.03 | 0.01 |
> |:-:|:-:|:-:|:-:|:-:|
> | 4-bit([0-th, 80-th]) | 69.45 | 68.63 | 67.69 | 65.90 |
> | 6-bit([0-th, 80-th]) | 70.22 | 68.87 | 67.15 | 66.10 |
> | 8-bit([0-th, 80-th]) | 69.96 | 68.66 | 66.75 | 64.99 |
> | FP([0-th, 160-th]) | 70.45 | 69.53 | 67.47 | 64.50 |
>
>
> Analysis: We can observe that as the learning rate sufficiently reduces for the first stage of [0-th, 80-th], adopting a lower precision for this stage will lead to a higher accuracy than training with full precision. In particular, with the standard initial learning rate 0.1, full precision training achieves a 1.00% / 0.70% higher accuracy than the 4-bit one on ResNet-38 / 74, respectively; whereas as the initial learning rate decreases, this accuracy gap gradually narrows and then reverses, e.g., when the initial learning rate becomes 1e-2, training with [0-th, 80-th] of 4-bit achieves a 1.40% / 1.57% higher accuracy than the full precision ones.
>
> This set of experiments show that (1) when the initial learning rate is low, training with lower initial precisions consistently lead to a better accuracy than training with full precision, indicating that lowering the precision introduces a similar effect of favoring exploration as that of a high learning rate; and (2) although a low precision can alleviate the accuracy drop caused by a low learning rate, a high learning rate is in general necessary to maximize the accuracy.
>
> In addition, while we leave the theoretical proof of CPT as our future work, there have been works that show how noise helps DNN training theoretically or empirically, e.g., “Noise injection: Theoretical prospects” (Y. Grandvalet, Neural Computation’97) and “Adding gradient noise improves learning for very deep networks” (A. Neelakantan, ICLR’16).
>
> **2. The optimizer for transformers.**
>
> We use the Adam optimizer for training transformers with the settings following FAIRSEQ (M. Ott, NAACL’19).
>
> **3. Static precision first and then CPT during training.**
>
> Following your suggestion, we conduct this set of experiments. Specifically, we train ResNet-38 / 74 on CIFAR-100 for 160 epochs, considering two CPT precision ranges with different starting epochs to enable CPT as shown in the tables below. We consistently observe that (1) an early starting epoch of CPT leads to a better accuracy on all the considered precision ranges and networks, and (2) even CPT with a late starting epoch still achieves a notable better accuracy than training without CPT.
>
> ***ResNet-74 on CIFAR-100:***
>
> | Starting Epoch | 0 | 60 | 80 | 100 | 120 | 160 (i.e., No CPT) |
> |:-:|:-:|:-:|:-:|:-:|:-:|:-:|
> | FW(3,8) / BW8 | 72.35 | 70.55 | 69.44 | 69.19 | 68.96 | 64.2 |
> | FW(3,6) / BW6 | 70.83 | 70.32 | 69.25 | 68.72 | 68.56 | 64.2 |
>
>
> ***ResNet-38 on CIFAR-100:***
>
> | Starting Epoch | 0 | 60 | 80 | 100 | 120 | 160 (i.e., No CPT) |
> |:-:|:-:|:-:|:-:|:-:|:-:|:-:|
> | FW(3,8) / BW8 | 70.24 | 69.60 | 68.37 | 68.04 | 67.65 | 62.3 |
> | FW(3,6) / BW6 | 69.30 | 68.65 | 68.00 | 67.83 | 67.85 | 62.3 |

---

### Decision · Program_Chairs · 2021-01-07
**Final Decision**

**Decision:**

Accept (Spotlight)

**Comment:**

All of the reviewers are impressed by this paper's empirical results and they agree that this is a good paper and should be accepted. Some questions about the theoretical justification of the proposed method and its potential practical impact remain open, but the empirical results are impressive and can result in more research in understanding Cyclic Precision Training (CPT) and improving quantized training of neural nets. I suggest acceptance as a spotlight presentation.